# Investigation of the status of rest facilities at industrial sites and awareness of relevant laws and regulations of South Korea

**Yeon-Hee Jeong**[1], **Woo-Je Lee**[1], **Ki-Youn Kim**[1,2]*

1 Graduate School of Safety Engineering, Seoul National University of Science and Technology, Seoul, Rep. of Korea, 2 Department of Safety Engineering, Seoul National University of Science and Technology, Seoul, Rep. of Korea

* kky5@seoultech.ac.kr

**Data Availability Statement:** All relevant data are within the manuscript and its Supporting Information files.

## Abstract

South Korea has faced many social issues due to long working hours, lack of rest areas, and poor rest facility environments for cleaners, security guards, department store workers, etc. Discussions have been ongoing about mandating the installation of rest facilities. From August 18, 2022, Article 128–2 of the Occupational Safety and Health Act, concerning the installation of rest facilities, was enforced. Consequently, employers in all industries are required to install rest facilities, and laws have been established to ensure these facilities meet certain standards. Accordingly, this study investigated the current status of rest facility installations and the awareness of the law's enactment in Korean industrial sites. The results, analyzed by gender, age, managerial status, industry, and size of the business, indicated that younger people were less satisfied with the rest facilities. Managers were more knowledgeable about the legal regulations than workers. In the service industry, compared to other industries, smaller businesses were less likely to have rest facilities and were less aware of the legal regulations. The results of this study are expected to be used as basic data to help establish the rest facility installation laws in Korea.

## 1. Introduction

The working hours of South Korean workers are among the highest, ranking 5th among 38 OECD countries in 2021. The top four countries with higher working hours were Mexico, Costa Rica, Colombia, and Chile, all in Latin America. Compared to Germany, which had the lowest number of working hours, South Korea worked 566 more hours annually, totaling 1.4 times the amount of working hours.

According to the research by Park Sung-jin et al. [1], it has been found that long working hours and shift workers who not getting sufficient rest are highly associated with health problems. Research by Isa Halim et al. [2] suggests that workers who stand for long periods in workplace have risk of various health impacts, and that preventive measures should include not only ergonomic solutions like seating but also rest periods. A NASA research report [3] on the sleep quantity and quality in the sleep spaces for aircraft crew members showed that the

**Funding:** The author(s) received no specific funding for this work.

**Competing interests:** The authors have declared that no competing interests exist.

type of aircraft influenced positive evaluations of rest spaces, and crew members who were satisfied with their rest had improved work performance and caution.

Rest periods are an important factor in enhancing the job performance of employees. Therefore, it is necessary to provide spaces and facilities that employees to have adequate rest periods [4]. A study on rest spaces for nurses in China [5] found that well-designed rest spaces positively affect nurses and increase their satisfaction with their work environment. Research on the satisfaction with welfare facilities on construction sites [6] indicated that factors such as dining facilities, clothes changing rooms, and rest facilities have a 60.9% impact. A study on the work and rest spaces for medical staff [7] stated that suitable interiors can create a healthy, happy, safe, and comfortable environment, leading to greater satisfaction among medical staff. Factors that affect employee satisfaction and the organization include clean offices and rest space [8]. Rest at work plays an important role in building positive relationships and recharging [9].

Research on workplace rest facilities has also been conducted in South Korea. According to the study by Kim Jong-min and Kang Kyung-sik [10], the establishment of rest facilities at construction sites affects the reduction of accidents, with observation variables such as the scale of rest areas, space provision, a variety of amenities, the installation of clothes changing rooms and showers, the provision of facilities for sleep, and the proximity of accommodation and rest facilities all influencing the decrease in accidents. Moreover, according to the research by Son Chang-baek [11], although there were some preliminary studies on the satisfaction of rest facilities for workers at construction sites, there were no surveys on the status and satisfaction of rest facilities across various industries. Korea Occupational Safety and Health Association (KOSHA) [12] conducted a study to review the size and type of businesses subject to sanctions, as well as the area, location, temperature, humidity, and management standards of rest facilities, in order to prepare subordinate legislation following the new legal responsibility for employers to install rest facilities.

Examining other country's law related to rest facilities, Türkiye's Occupational Health and Safety Law [13] defines a workplace as a facility that includes rest space. India's labor laws [14] specify shelters for food and rest that must be provided and maintained for workers' use. The UK guidelines for welfare provisions for workers [15] include rest facilities as part of the welfare facilities. In Australia's law, provide rest facilities for truck drivers, to manage fatigue and comply with driving time regulations [16].

In line with the international trend, South Korea's Occupational Safety and Health Law has been amended, requiring all workplaces to install rest facilities regardless of the type and size of the business from August 18, 2022, according to Article 128-2(1). However, since not enough time has passed since the enactment of the law, its provisions have not yet been fully established.

Therefore, this study aims to provide foundational data for the future establishment of legal systems and the improvement of workers' working environments by surveying the current installation and satisfaction levels of rest facilities in Korean industrial sites, as well as the awareness of related legal regulations.

## 2. Method

### 2.1 Subject

A survey was conducted from April 23 to June 30, 2023, targeting general workers and managers (site managers, supervisor, safety managers, health managers, etc.) employed in construction, manufacturing, and other service industries.

This study received prior approval from the Seoul National University of Science and Technology Institutional Review Board (approval number: 2023-0005-01). The questionnaire was distributed online via Google Forms after obtaining consent from participants who were informed about the purpose of the research, survey methods, and the use of the survey data.

The required number of participants for the research was calculated using the G*Power program. A total of 200 samples were needed when applying ANOVA statistical analysis, effect size f = 0.25, aerr prob = 0.05, and Power = 0.8. Taking into account the dropout rate, 227 survey responses were ultimately collected for this study.

## 2.2 Survey item

The questionnaire consisted of a total of 37 questions, including 5 questions related to general characteristics, 23 questions on the presence of rest facilities and compliance with relevant standards, and 9 questions related to the enactment of laws and perceptions of rest facilities (See Table 1).

## 2.3 Data analysis

All statistical analyses were conducted using SPSS ver.26.0 (IBM, USA). Frequency analysis was conducted to understand the proportions of each response, and independent t-tests and one-way ANOVA were conducted to investigate differences in responses according to general characteristics. In cases where significant differences were found in the one-way ANOVA, post-hoc analyses were performed using LSD, S-N-K, Scheffe, and Duncan methods. Reliability analysis was conducted for items (total of 26) that could be tested resulting in Cronbach's α = 0.781, indicating that the survey results are reliable.

## 3. Results

### 3.1 General and work-related characteristics

As shown in Table 2, among the 227 participants in the survey, 76.7% were male and 23.3% were female. The distribution across age groups from the 20s to the 50s was relatively even, ranging from 15.4% to 38.8%. However, only one person, accounting for 0.4%, was over the age of 60.

In terms of industry distribution, manufacturing accounted for 33.9%, construction 32.6%, and other services 33.5%. The proportion of managers such as safety managers, health managers, and supervisor was 75.8%, while general workers was 24.2%. The proportion by business size was highest for workplaces with more than 300 employees (or construction sites with a contract amount of more than 12 billion), accounting for 60.8% of the total.

### 3.2 Rest facility installation and management standards

Out of the total 227 respondents, 205, accounting for over 90%, indicated that rest facilities were available. As shown in Tables 3 and 4, over 50% of the respondents stated that the available rest facilities met or exceeded the standards in terms of area, height, maintenance of temperature and humidity, installation of lighting, location in a safe place, provision of water and chairs, and designation of a manager. However, only about 48.5% responded that the rest facilities were segregated by gender, indicating that this aspect does not meet the standards.

**3.2–1 Reason of rest installation uninstalled.** Of the 27 respondents who indicated that no rest facilities were available, a survey was conducted to determine the reasons for this absence. The most common response, from 18 respondents (33%), was the lack of a suitable location, followed by cost issues for 16 respondents (30%), lack of awareness by the employer

**Table 1. Survey contents.**

| | Label | Question | Count |
|---|---|---|---|
| *Genenal characteristics* | - | **sex, age, industry, job title, size** | **5** |
| *Rset facility related (environment)* | Q2 | Are rest facilities installed? | 23 |
| | Q2-1 | Why not installed the rest facility?(no responders only) | |
| | Q2-2 | Why not installed the rest facility?(total investigator) | |
| | Q3 | Is it segregated into male and female? | |
| | Q4 | Is it more than 6square meters? | |
| | Q5 | Is the area suitable for simultaneous use? | |
| | Q6 | Are they taller than 2.1 meters? | |
| | Q7 | Is there air conditioning/heating? | |
| | Q7-1 | Why is there no air conditioning? | |
| | Q8 | Is it in a safe place from fire and explosion hazard? | |
| | Q8-1 | Why is it installed in a fire and explosion hazard area? | |
| | Q9 | Is it installed in a safe place from noise and dust? | |
| | Q9-2 | Why is it installed in a noise and dust? | |
| | Q10 | Is water provided? | |
| | Q11 | Are there chairs? | |
| | Q12 | Is there a rest facility manager? | |
| | Q13 | Are you using the rest facilities? | |
| | Q13-1 | Why not use the rest facility? | |
| | Q14 | Is it a private resting facility? | |
| | Q15 | Are you maintaining the right temperature? | |
| | Q16 | Are you maintaining proper humidity? | |
| | Q17 | Is the lighting adequate? | |
| | Q20 | Are you satisfied with the currently installed rest facilities? | |
| *Rset facility law related (awareness)* | Q1 | Do you know that the rest facility installation law was enacted? | 9 |
| | Q18 | Do you think rest facility manager is needed? | |
| | Q19 | Do you think rest facilities are essential? | |
| | Q21 | Do you think rest facilities are helpful in preventing safty accidents? | |
| | Q22 | Do you think rest facilities are helpful for improving health? | |
| | Q23 | Do you think a business with less than 20 employees needs financial support? | |
| | Q24 | Do you think a penalty of 15million won for not installing rest facilities is appropriate? | |
| | Q25 | Do you think a penalty of 5million won is appropriate when the rest facility is below the standard? | |
| | Q26 | Do you think the enactment of the law will help improve rest facilities? | |

for 14 respondents (26%), lack of knowledge about relevant laws for 4 respondents (7%), and 2 respondents (4%) who did not feel the necessity for such facilities.

Additionally, when all respondents were asked about the reasons for the inability to install rest facilities in workplaces in Korea, 222 responded. The reasons were in the order of lack of space for 96 respondents (43%), lack of employer's will for 68 respondents (31%), budget constraints for 53 respondents (24%), 3 respondents (1%) who did not see the need for rest facilities, and 2 respondents (1%) citing difficulties in construction.

**3.2–2 Reason of rest installation no use.** Out of the 205 respondents who responded having rest facilities, 48 stated they did not use them despite their availability. The most common reason, cited by 19 respondents (39.6%), was 'not feeling a significant need for rest facilities'. Other reasons included the space being too small and feeling self-conscious under the manager's watchful eye, reported by 10 respondents (20.8%) and 7 respondents (14.6%) respectively. Excluding those who did not feel the need for rest facilities, the most common reason for not

**Table 2. General characteristics and work-related characteristics of the study subjects.**

| Classification | Category | People | % |
|---|---|---:|---:|
| Sex | Man | 174 | 76.7 |
| | Woman | 53 | 23.3 |
| Age | 20~29 | 35 | 15.4 |
| | 30~39 | 56 | 24.7 |
| | 40~49 | 88 | 38.8 |
| | Over 50 | 48 | 21.1 |
| Industry | Manufacturing business | 77 | 33.9 |
| | Construction industry | 74 | 32.6 |
| | Service industry | 76 | 33.5 |
| Job title | Manager | 172 | 75.8 |
| | Worker | 55 | 24.2 |
| Scale | <20 peaple | 39 | 17.2 |
| | 20~50 peaple | 15 | 6.6 |
| | 51~300 peaple | 35 | 15.4 |
| | >300 peaple | 138 | 60.8 |

using them was the limited space available. This mirrors the reason cited for the inability to install rest facilities, which was 'lack of appropriate space or location'.

### 3.3 Independent t-test results by sex

The results of the independent t-test (shown in Table 5) by gender showed that male were statistically significantly more aware of the enactment of the law mandating the installation of rest facilities than female (p = 0.043).

### 3.4 One-way ANOVA results by age

The results of the one-way ANOVA based on age indicated that individuals in their 30s responded statistically significantly more often than those over 50s that the lighting in the rest facilities was inadequate (p = 0.005). Additionally, those in their 30s were statistically

**Table 3. Frequency analysis–Q1~Q14.**

| NO. | yes(%) | no(%) | total(%) |
|:---:|:---:|:---:|:---:|
| Q1 | 168(74) | 59(26) | 227(100) |
| Q2 | 205(90.3) | 22(9.7) | 227(100) |
| Q3 | 110(48.5) | 94(41.4) | 204(89.9) |
| Q4 | 190(83.7) | 12(5.3) | 202(89) |
| Q5 | 149(65.6) | 53(23.3) | 202(89) |
| Q6 | 192(84.6) | 11(4.8) | 203(89.4) |
| Q7 | 184(81.1) | 20(8.8) | 204(89.9) |
| Q8 | 188(82.8) | 16(7) | 204(89.9) |
| Q9 | 170(74.9) | 33(14.5) | 203(89.4) |
| Q10 | 182(80.2) | 19(8.4) | 201(88.5) |
| Q11 | 195(85.9) | 8(3.5) | 203(89.4) |
| Q12 | 157(69.2) | 42(18.5) | 199(87.7) |
| Q13 | 173(76.2) | 29(12.8) | 202(89) |
| Q14 | 157(69.2) | 43(18.9) | 200(88.1) |

**Table 4. Frequency analysis—Q15~Q26.**

|  | Storngly yes(%) | Yes(%) | Normal(%) | No(%) | Absolutely no(%) | total(%) |
|---|---|---|---|---|---|---|
| Q15 | 61(26.9) | 90(39.6) | 46(0.3) | 5(2.2) | 0(0) | 202(89) |
| Q16 | 40(17.6) | 69(30.4) | 47(20.7) | 39(17.2) | 7(3.1) | 202(89) |
| Q17 | 59(26) | 97(42.7) | 45(19.8) | 2(0.9) | 0(0) | 203(89.4) |
| Q18 | 62(27.3) | 88(38.8) | 36(15.9) | 8(3.5) | 1(0.4) | 195(85.9) |
| Q19 | 113(49.8) | 87(38.3) | 18(7.9) | 5(2.2) | 0(0) | 223(98.2) |
| Q20 | 41(18.1) | 104(45.8) | 60(26.4) | 6(2.6) | 4(1.8) | 215(94.7) |
| Q21 | 57(25.1) | 116(51.1) | 42(18.5) | 11(4.8) | 1(0.4) | 227(100) |
| Q22 | 81(35.7) | 109(48) | 34(15) | 2(0.9) | 1(0.4) | 227(100) |
| Q23 | 80(35.2) | 92(40.5) | 36(15.9) | 13(5.7) | 5(2.2) | 226(99.6) |
| Q24 | 27(11.9) | 58(25.6) | 71(31.3) | 52(22.9) | 19(8.4) | 227(100) |
| Q25 | 24(10.6) | 60(26.4) | 78(34.4) | 48(21.1) | 17(7.5) | 227(100) |

significantly more likely than those over 50s to respond that the enactment of the law did not help in the installation or improvement of rest facilities (p = 0.041). People in their 20s were statistically significantly more likely than those in their 30s and older to respond that the rest facilities were not safe from dust and noise (p = 0.000).

### 3.5 One-way ANOVA results by industry

The results of the one-way ANOVA and post-hoc analysis by industry type (manufacturing, construction, and other services), as shown in Tables 6 and 7, revealed that other services were statistically significantly less aware of the enactment of the law compared to other industries (p = 0.0000), tended to have areas less than the standard of 6m$^2$ (p = 0.02), were less likely to have designated managers (p = 0.004), and were less likely to have dedicated rest facilities (p = 0.009).

Furthermore, in the construction industry, there was a statistically significant tendency to fail in segregating rest facilities by gender compared to other industries (p = 0.021). In manufacturing, there was a statistically significant tendency to maintain higher humidity compared to other industries (p = 0.018)

### 3.6 Independent sample t-test results by job title

The results of the independent t-test regarding job position showed that managers were statistically significantly more aware of the enactment of the law concerning rest facilities compared to workers (p = 0.000). They also tended to respond more affirmatively regarding the installation of rest facilities (p = 0.009) and compliance with legal standards for rest facilities. Regarding the appropriateness of imposing fines for not installing rest facilities, workers were statistically significantly more likely than managers to respond that it was appropriate (p = 0.012), as shown in Table 4.

### 3.7 One-way ANOVA results by company size

The results of the one-way ANOVA and post-hoc analysis by business size, as shown in Tables 8 and 9, indicated that workplaces with fewer than 20 employees were statistically significantly less likely to be aware of the enactment of the rest facilities law compared to other sizes (p = 0.000). When comparing groups of more than 50 employees, between 20 and 50 employees, and fewer than 20 employees, the smaller the size of the workplace, the more statistically significant was the tendency to respond that there were no rest facilities (p = 0.000).

**Table 5. Result by position (independent t-test).**

| Question | Answer | Manager | Worker | t | p-value (two side test) |
|---|---|---|---|---|---|
| Q1 | yes | 143 | 25 | -5.123 | 0.000*** |
| | no | 29 | 30 | | |
| Q2 | yes | 162 | 43 | -2.713 | 0.009** |
| | no | 10 | 12 | | |
| Q3 | yes | 86 | 24 | 0.279 | 0.781 |
| | no | 75 | 19 | | |
| Q4 | yes | 154 | 36 | -1.859 | 0.069 |
| | no | 6 | 6 | | |
| Q5 | yes | 117 | 32 | 0.110 | 0.913 |
| | no | 42 | 11 | | |
| Q6 | yes | 152 | 40 | -0.506 | 0.613 |
| | no | 8 | 3 | | |
| Q7 | yes | 114 | 40 | 0.699 | 0.485 |
| | no | 17 | 3 | | |
| Q8 | yes | 152 | 36 | -1.788 | 0.080 |
| | no | 9 | 7 | | |
| Q9 | yes | 134 | 36 | -0.005 | 0.996 |
| | no | 26 | 7 | | |
| Q10 | yes | 145 | 37 | -0.608 | 0.544 |
| | no | 14 | 5 | | |
| Q11 | yes | 153 | 42 | 0.611 | 0.542 |
| | no | 7 | 1 | | |
| Q12 | yes | 127 | 30 | -1.225 | 0.226 |
| | no | 30 | 12 | | |
| Q13 | yes | 136 | 37 | 0.085 | 0.933 |
| | no | 23 | 6 | | |
| Q14 | yes | 127 | 30 | -1.157 | 0.252 |
| | no | 31 | 12 | | |
| Q15 | 1 | 52 | 9 | -1.100 | 0.273 |
| | 2 | 69 | 21 | | |
| | 3 | 35 | 11 | | |
| | 4 | 4 | 1 | | |
| | 5 | 0 | 0 | | |
| Q16 | 1 | 36 | 4 | -1.301 | 0.195 |
| | 2 | 54 | 15 | | |
| | 3 | 33 | 14 | | |
| | 4 | 30 | 9 | | |
| | 5 | 6 | 1 | | |
| Q17 | 1 | 53 | 6 | -2.366 | 0.019* |
| | 2 | 74 | 23 | | |
| | 3 | 31 | 14 | | |
| | 4 | 2 | 0 | | |
| | 5 | 0 | 0 | | |
| Q18 | 1 | 49 | 13 | -1.343 | 0.181 |
| | 2 | 73 | 15 | | |
| | 3 | 26 | 10 | | |
| | 4 | 4 | 4 | | |

*(Continued)*

**Table 5.** (Continued)

| Question | Answer | Manager | Worker | t | p-value (two side test) |
|---|---|---|---|---|---|
| | 5 | 1 | 0 | | |
| Q19 | 1 | 87 | 26 | -0.842 | 0.401 |
| | 2 | 67 | 20 | | |
| | 3 | 9 | 9 | | |
| | 4 | 5 | 0 | | |
| | 5 | 0 | 0 | | |
| Q20 | 1 | 34 | 7 | -2.509 | 0.015* |
| | 2 | 84 | 20 | | |
| | 3 | 45 | 15 | | |
| | 4 | 2 | 4 | | |
| | 5 | 1 | 3 | | |
| Q21 | 1 | 47 | 10 | -1.246 | 0.214 |
| | 2 | 86 | 30 | | |
| | 3 | 31 | 11 | | |
| | 4 | 7 | 4 | | |
| | 5 | 1 | 0 | | |
| Q22 | 1 | 61 | 20 | 0.064 | 0.949 |
| | 2 | 83 | 26 | | |
| | 3 | 26 | 8 | | |
| | 4 | 1 | 1 | | |
| | 5 | 1 | 0 | | |
| Q23 | 1 | 63 | 17 | -1.843 | 0.069 |
| | 2 | 73 | 19 | | |
| | 3 | 25 | 11 | | |
| | 4 | 7 | 6 | | |
| | 5 | 3 | 2 | | |
| Q24 | 1 | 17 | 10 | 1.874 | 0.062 |
| | 2 | 41 | 17 | | |
| | 3 | 59 | 12 | | |
| | 4 | 38 | 14 | | |
| | 5 | 17 | 2 | | |
| Q25 | 1 | 16 | 8 | 2.543 | 0.012* |
| | 2 | 39 | 21 | | |
| | 3 | 63 | 15 | | |
| | 4 | 39 | 9 | | |
| | 5 | 15 | 2 | | |
| Q26 | 1 | 39 | 13 | 0.884 | 0.378 |
| | 2 | 83 | 29 | | |
| | 3 | 39 | 11 | | |
| | 4 | 6 | 2 | | |
| | 5 | 5 | 0 | | |

* p<0.05

** P<0.01

*** p<0.001

1) very much yes / 2) yes / 3) nomal / 4) no / 5) very much no

**Table 6. Result by industry(One-way ANOVA).**

|  | answer | Manufacturing business | Construction industry | Service industry | F | Significance probability |
|---|---|---|---|---|---|---|
| Q1 | yes | 63 | 64 | 41 | 13.446 | 0.000*** |
|  | no | 14 | 10 | 35 |  |  |
| Q2 | yes | 72 | 68 | 65 | 1.549 | 0.215 |
|  | no | 5 | 6 | 11 |  |  |
| Q3 | yes | 47 | 28 | 35 | 3.945 | 0.021* |
|  | no | 25 | 39 | 30 |  |  |
| Q4 | yes | 70 | 64 | 56 | 3.994 | 0.02* |
|  | no | 1 | 3 | 8 |  |  |
| Q5 | yes | 52 | 50 | 47 | 0.112 | 0.894 |
|  | no | 20 | 16 | 17 |  |  |
| Q6 | yes | 65 | 66 | 61 | 1.682 | 0.189 |
|  | no | 6 | 1 | 4 |  |  |
| Q7 | yes | 66 | 57 | 61 | 1.572 | 0.210 |
|  | no | 6 | 10 | 4 |  |  |
| Q8 | yes | 62 | 66 | 60 | 3.773 | 0.025* |
|  | no | 10 | 1 | 5 |  |  |
| Q9 | yes | 60 | 52 | 58 | 2.054 | 0.131 |
|  | no | 12 | 15 | 6 |  |  |
| Q10 | yes | 65 | 61 | 56 | 0.237 | 0.789 |
|  | no | 7 | 5 | 7 |  |  |
| Q11 | yes | 68 | 64 | 63 | 0.380 | 0.684 |
|  | no | 4 | 2 | 2 |  |  |
| Q12 | yes | 62 | 53 | 42 | 5.722 | 0.004** |
|  | no | 8 | 12 | 22 |  |  |
| Q13 | yes | 62 | 54 | 57 | 0.580 | 0.561 |
|  | no | 9 | 12 | 8 |  |  |
| Q14 | yes | 61 | 53 | 43 | 4.817 | 0.009** |
|  | no | 9 | 12 | 22 |  |  |

For workplaces with fewer than 20 employees, the response that the rest facilities did not meet the area standard of $6m^2$ or more (p = 0.000) and were not dedicated rest facilities (p = 0.019) was statistically significantly higher compared to other sizes. Conversely, workplaces with fewer than 20 employees were found to be without a designated person in charge

**Table 7. Post analysis by industry.**

| Question | Post-hoc analysis | | | |
|---|---|---|---|---|
|  | LSD | S-N-K | Duncan | Scheffe |
| Q1 | a*, b** < c*** | a, b < c | a, b < c | a, b < c |
| Q2 | a < b | a < b | a < b | a < b |
| Q4 | a < c | a, b < c | a, b < c | a < c |
| Q8 | b < a | b < a | b < a | b < a |
| Q12 | a, b < c | a, b < c | a, b < c | a < c |
| Q14 | a, b < c | a, b < c | a, b < c | a < c |
| Q16 | a < b, c | a < b, c | a< b, c | a < c |

a: Manufacturing business / b: Construction industry / c: Service industry

**Table 8. Result by company size(One-way ANOVA).**

| | answer | <20 person = 2billion | 20~50person = 2~5billion | 51~300person = 5~12billion | >300person = 12billion | F | Significance probability |
|---|---|---|---|---|---|---|---|
| Q1 | yes | 16 | 10 | 28 | 114 | 10.645 | 0.000*** |
| | no | 23 | 5 | 7 | 24 | | |
| Q2 | yes | 25 | 12 | 34 | 134 | 16.662 | 0.000*** |
| | no | 14 | 3 | 1 | 4 | | |
| Q3 | yes | 10 | 8 | 14 | 78 | 2.073 | 0.105 |
| | no | 15 | 4 | 20 | 55 | | |
| Q4 | yes | 19 | 11 | 30 | 130 | 7.224 | 0.000*** |
| | no | 6 | 1 | 3 | 2 | | |
| Q5 | yes | 17 | 9 | 24 | 99 | 0.130 | 0.942 |
| | no | 7 | 3 | 10 | 33 | | |
| Q6 | yes | 22 | 11 | 30 | 129 | 1.568 | 0.198 |
| | no | 3 | 1 | 3 | 4 | | |
| Q7 | yes | 23 | 12 | 30 | 119 | 0.535 | 0.659 |
| | no | 2 | 0 | 4 | 14 | | |
| Q8 | yes | 20 | 11 | 32 | 125 | 1.990 | 0.117 |
| | no | 5 | 1 | 2 | 8 | | |
| Q9 | yes | 18 | 11 | 27 | 114 | 1.408 | 0.242 |
| | no | 7 | 1 | 7 | 18 | | |
| Q10 | yes | 20 | 9 | 31 | 122 | 1.963 | 0.121 |
| | no | 4 | 3 | 2 | 10 | | |
| Q11 | yes | 24 | 10 | 33 | 128 | 1.852 | 0.139 |
| | no | 1 | 2 | 1 | 4 | | |
| Q12 | yes | 11 | 8 | 29 | 109 | 7.303 | 0.000*** |
| | no | 13 | 4 | 5 | 20 | | |
| Q13 | yes | 21 | 9 | 26 | 117 | 1.124 | 0.340 |
| | no | 4 | 3 | 7 | 15 | | |
| Q14 | yes | 13 | 10 | 29 | 105 | 3.407 | 0.019* |
| | no | 11 | 2 | 5 | 25 | | |
| Q15 | 1 | 4 | 4 | 10 | 43 | 1.310 | 0.272 |
| | 2 | 12 | 7 | 17 | 54 | | |
| | 3 | 8 | 1 | 5 | 32 | | |
| | 4 | 1 | 0 | 2 | 2 | | |
| | 5 | 0 | 0 | 0 | 3 | | |
| Q16 | 1 | 1 | 1 | 6 | 32 | 1.556 | 0.201 |
| | 2 | 9 | 6 | 11 | 43 | | |
| | 3 | 9 | 1 | 5 | 32 | | |
| | 4 | 5 | 3 | 12 | 19 | | |
| | 5 | 1 | 1 | 0 | 5 | | |
| Q17 | 1 | 4 | 2 | 12 | 41 | 1.347 | 0.260 |
| | 2 | 14 | 5 | 13 | 65 | | |
| | 3 | 7 | 5 | 8 | 25 | | |
| | 4 | 0 | 0 | 1 | 1 | | |
| | 5 | 0 | 0 | 0 | 2 | | |
| Q18 | 1 | 0 | 4 | 12 | 46 | 7.972 | 0.000*** |
| | 2 | 11 | 4 | 12 | 61 | | |
| | 3 | 9 | 4 | 8 | 15 | | |
| | 4 | 4 | 0 | 0 | 4 | | |

(*Continued*)

**Table 8.** (Continued)

| | answer | <20 person = 2billion | 20~50person = 2~5billion | 51~300person = 5~12billion | >300person = 12billion | F | Significance probability |
|---|---|---|---|---|---|---|---|
| | 5 | 0 | 0 | 0 | 1 | | |
| Q19 | 1 | 12 | 8 | 17 | 76 | 3.880 | 0.000*** |
| | 2 | 17 | 6 | 16 | 48 | | |
| | 3 | 9 | 1 | 0 | 8 | | |
| | 4 | 1 | 0 | 1 | 3 | | |
| | 5 | 0 | 0 | 1 | 3 | | |
| Q20 | 1 | 3 | 1 | 7 | 30 | 2.452 | 0.064 |
| | 2 | 15 | 10 | 17 | 62 | | |
| | 3 | 12 | 1 | 10 | 37 | | |
| | 4 | 3 | 1 | 1 | 1 | | |
| | 5 | 1 | 1 | 0 | 2 | | |
| Q21 | 1 | 6 | 1 | 12 | 38 | 1.740 | 0.160 |
| | 2 | 20 | 11 | 17 | 68 | | |
| | 3 | 10 | 2 | 5 | 25 | | |
| | 4 | 3 | 1 | 1 | 6 | | |
| | 5 | 0 | 0 | 0 | 1 | | |
| Q22 | 1 | 12 | 6 | 14 | 49 | 0.257 | 0.856 |
| | 2 | 19 | 7 | 16 | 67 | | |
| | 3 | 8 | 2 | 4 | 20 | | |
| | 4 | 0 | 0 | 1 | 1 | | |
| | 5 | 0 | 0 | 0 | 1 | | |
| Q23 | 1 | 17 | 5 | 14 | 44 | 0.661 | 0.577 |
| | 2 | 12 | 6 | 14 | 60 | | |
| | 3 | 6 | 1 | 6 | 23 | | |
| | 4 | 4 | 2 | 1 | 6 | | |
| | 5 | 0 | 1 | 0 | 4 | | |
| Q24 | 1 | 5 | 2 | 4 | 16 | 0.248 | 0.863 |
| | 2 | 10 | 5 | 10 | 33 | | |
| | 3 | 10 | 4 | 10 | 47 | | |
| | 4 | 10 | 4 | 8 | 30 | | |
| | 5 | 4 | 0 | 3 | 12 | | |
| Q25 | 1 | 2 | 2 | 3 | 17 | 0.810 | 0.490 |
| | 2 | 13 | 6 | 7 | 34 | | |
| | 3 | 14 | 4 | 13 | 47 | | |
| | 4 | 8 | 3 | 9 | 28 | | |
| | 5 | 2 | 0 | 3 | 12 | | |
| Q26 | 1 | 8 | 4 | 11 | 29 | 0.145 | 0.933 |
| | 2 | 19 | 7 | 12 | 74 | | |
| | 3 | 10 | 4 | 8 | 28 | | |
| | 4 | 2 | 0 | 3 | 3 | | |
| | 5 | 0 | 0 | 1 | 4 | | |

* p<0.05

** P<0.01

*** p<0.001

1) Storongly yes / 2) Yes / 3) Normal / 4) No / 5) Absolutely no

**Table 9. Post analysis by company size.**

| Question | Post-hoc analysis | | | |
|---|---|---|---|---|
| | LSD | S-N-K | Duncan | Scheffe |
| Q1 | b, c, d < a | b, c, d < a | b, c, d < a | c, d < a |
| Q2 | c, d < a, b | c, d < b < a | c, d < b < a | c, d < a |
| Q4 | c, d < a | b, c, d < a | b, c, d < a | d < a |
| Q12 | c, d < a | c, d < a | c, d < a | c, d < a |
| Q14 | b, c, d < a | b, c, d < a | b, c, d < a | n/a |
| Q18 | b, c, d < a | b, c, d < a | b, c, d < a | b, c, d < a |

A: <20 person / b: 20~50 person / c: 50~300 person / d: > 300person

of the rest facilities (p = 0.000) and also felt a higher necessity for a person in charge of rest facilities (p = 0.000) and for the rest facilities themselves (p = 0.000) compared to workplaces with more than 50 employees.

## 3.8 Awareness of relevant laws and regulations

Table 10 presents the results of a survey on the awareness of the enactment of the law related to rest facilities. Regarding the necessity of rest facilities, out of 223 respondents, about 89.7% (200 respondents) answered that rest facilities are necessary. When asked about their overall satisfaction with the current rest facilities, 145 respondents (67.5%) expressed satisfaction.

Concerning whether rest facilities help in preventing safety accidents and in health management, respectively, 173 respondents (76.2%) and 190 respondents (83.7%) answered positively (very much so, yes). In response to whether government financial support is needed for establishing rest facilities in workplaces with fewer than 20 employees (or construction sites with less than 200 million in contract amount), 172 respondents (76.1%) indicated that it is necessary (very much so, yes).

On the other hand, regarding the appropriateness of the fines for not installing rest facilities or for non-compliance with the rest facility standards, only 37.5% and 37.1% of all respondents, respectively, considered the fines appropriate (very much so, yes), indicating a higher proportion of negative responses.

When asked if the enactment of the related law has helped in the installation and improvement of rest facilities, 164 respondents (72.2%) responded positively that it has been helpful (very much so, yes).

**Table 10. Law-related awareness results number of respondents (%).**

| | very much yes | yes | normal | no | very much no |
|---|---|---|---|---|---|
| 1) Necessity of rest facilities | 113(50.7) | 87(39) | 18(8.1) | 5(2.2) | 0(0) |
| 2) Satisfaction with rest facilities | 41(19.1) | 104(48.4) | 60(27.9) | 6(2.8) | 4(1.9) |
| 3) Contributing to the prevention of accidents | 57(25.1) | 116(51.1) | 42(18.5) | 11(4.8) | 1(0.4) |
| 4) Contributing to the maintenance of health | 81(35.7) | 109(48) | 34(15) | 2(0.9) | 1(0.4) |
| 5) Need for financial support | 80(35.4) | 92(40.7) | 36(15.9) | 13(5.8) | 5(2.2) |
| 6) Suitability of non-install rest facilities penalty | 27(11.9) | 58(25.6) | 71(31.3) | 52(22.9) | 19(8.4) |
| 7) Suitability of substand rest facilities penalty | 24(10.6) | 60(26.4) | 78(34.4) | 48(21.1) | 17(7.5) |
| 8) Degree of improvement since the enactment of the law | 52(22.9) | 112(49.3) | 50(22) | 8(3.5) | 5(2.2) |

Total number(%): 1) 223person(98.2%) / 2) 215 person (94.7%) / 3) 227 person (100%) / 4) 227 person (100%) / 5) 226 person (99.6%) / 6) 227 person (100%) / 7) 227 person (100%) / 8) 227 person (100%)

## 4. Discussion

This study examined the installation and compliance of rest facilities at industrial sites in Korea, as well as awareness of the enactment of Article 128–2, Paragraph 1 of the Occupational Safety and Health Act. Although the responses of managers and workers to the question of the existence of rest facilities were statistically significantly different (p = 0.009), a Mann-Whitney U test conducted among managers and workers in workplaces with fewer than 50 employees (where the ratio of managers to workers is similar) showed no significant difference (p = 0.772). Considering that in workplaces with more than 300 employees, there were 112 managers and 26 workers, and that larger workplaces tend to have rest facilities, the difference seems to be related to workplace size rather than the role of managers or workers. And also there are studies [17] suggesting that managerial staff tend to have higher job satisfaction compared to production workers.

Regarding the reasons for the lack of rest facilities, all survey respondents (including those without rest facilities) cited 'lack of a suitable place for rest facilities', 'lack of awareness by the employer', 'cost issues', and 'other (not feeling the necessity, lack of legal knowledge)' in that order. This reflects the challenges of small businesses in allocating space for rest facilities, the high male ratio in construction, and the priority of customer space in the service industry. Some studies founds that workers in the catering industry, compared to those in manufacturing or heavy industry, reported a significant prevalence of pain in specific body parts like the back, shoulders, neck, and wrists [18–20]. Other Studies indicate that service workers experience high job stress, poor physical work environments, inappropriate rest breaks, and a lack of rest space [21–23]. Kwon Soo Bin et al. [24] noted that childcare teachers struggle with securing rest time due to the lack of a dedicated rest space or the nature of their work. According to Isa Halim et al. [2], workers who stand for long periods in industrial settings are at risk of various health impacts and need both engineering solutions like chairs and rest periods. These findings suggest the need for improvement in rest facilities, especially in the service sector.

Lee Hu Yeon et al. [25] reported that men in small manufacturing businesses with high stress are more prone to fatigue symptoms, and Lee Seung Hyun et al. [26] found that workers in manufacturing businesses with fewer than 50 employees have a higher rate of musculoskeletal symptom awareness than those in larger corporations or medium-sized enterprises. According to Cho Eui Joon et al. [27], smaller-scale workplaces tend to have a higher incidence of occupational diseases. Research by Ahn Yeon Soon et al. [28] and Yum Yong Tae [29] showed that the approval rate for rest due to occupational diseases is high in manufacturing, following mining, with high rates for dust and noise-related approvals. Studies by Kim Heon et al. [30] and Park Jeong Duk et al. [31] indicated that exposure to commonly used industrial substances like organic solvents and heavy metals leads to various diseases, so preventing exposure is key to disease prevention. As these studies suggest, manufacturing (especially small-scale) needs rest facilities separated from the workspace.

In the construction industry, as shown in the study by Son Chang Baek et al. [11], satisfaction with the rest facilities installed at construction sites was very low. Even the sites of large construction companies within the top 20 in terms of construction ranking as of 2010 did not meet the area standards for rest facilities and often lacked heating, seating, or beverage facilities. Although Paik Shin Won et al. [32] found that higher age in construction site workers does not necessarily lead to more safety accidents, Lee Mi Ra et al. [33] reported that outdoor construction work leads to high job stress and depression due to poor working conditions, with significant differences observed with increasing age. Given these factors, including the prevalence of older workers and the physical stress due to outdoor work conditions, the installation of rest facilities in the construction industry is deemed essential for preventing safety accidents.

Following the legal amendment, the Korean Ministry of Employment and Labor has been imposing fines during site inspections for system stabilization and promoting the law. Financial support for small businesses is also being provided, suggesting that compliance with the installation and standards of rest facilities will improve gradually. However, the issue of space shortage in construction and small businesses, which prevents compliance with the law, requires urgent attention. As shown in the study by Cho Sung Hye et al. [34], 'workers in similar employment types (special employment type workers)' are still under limited protection according to the current Korean Industrial Safety and Health Act, and measures are needed for these unprotected workers.

Regarding the imposition of fines, the high rate of negative responses may be attributed to over 76% of the respondents being managers. Nevertheless, 89.7% responded that rest facilities are necessary, and 72.2% believed that the enactment of the law would help improve rest facilities, indicating a generally positive expectation of the mandatory enactment of the rest facility-related law.

In this study, the satisfaction with rest facilities showed no statistically significant differences by industry ($p = 0.248$) or size ($p = 0.064$), but managers were more satisfied than workers ($p = 0.015$). A study [35] on factors affecting workers' satisfaction with rest facilities, conducted with 906 workers from 150 businesses nationwide, found an average satisfaction score of 3.2 out of 5. Factors influencing satisfaction included younger age, being male, holding a position of assistant manager or higher, working in a business with 300 or more employees, having four or more rest facilities, having signs for rest facilities, believing that rest facilities improve work efficiency, experiencing lower fatigue, and frequent use of rest facilities. Therefore, when installing rest facilities, various factors should be considered to ensure worker satisfaction.

This study has several limitations. First, respondent and business information was collected anonymously to enhance response rate and reliability, leading to an uneven distribution of business sizes and ratios of managers to workers. Second, there was no field survey of rest facilities in the respondents' workplaces. Third, there was a low response rate from workers in other service industries like call center operators, cleaners, and security guards, who might have a high need for rest facilities.

Nevertheless, this research is the first to investigate the installation and compliance of rest facilities, and awareness of related legal regulations in Korean workplaces after the enactment of the rest facility-related law. The results of this study are expected to be used for increasing the installation rate of rest facilities, improving related systems, and serving as foundational data for future research in rest facility-related fields.

## 5. Conclusion

In Korea, due to long working hours and high job stress in certain occupations, the necessity of rest times and facilities was recognized, leading to a legal amendment that mandated the installation of rest facilities in all workplaces from August 18, 2022. Following this legislation, a survey on the installation status and compliance with standards of rest facilities in domestic workplaces, as well as awareness of the related laws, showed that, apart from men being more aware of the law's enactment than women, no other significant results were found related to different variables. In terms of age, younger respondents expressed lower satisfaction with rest facilities and reported non-compliance with standards.

In manufacturing, compared to other industries, responses indicating that rest facility standards were adequately met were higher. The construction industry reported difficulties in installing rest facilities due to site-specific issues like lack of space and electrical problems.

Other service industries, which experience high job stress, showed lower awareness of the law's enactment and significant differences from other industries in terms of lack of space, absence of managers, and not having dedicated rest facilities.

Managers were found to be more aware of the law and regulations compared to field workers and responded that rest facilities met installation standards. However, they tended to view the fines for not installing rest facilities as inappropriate.

Regarding business size, employees in smaller businesses were less aware of the enactment of the law, more likely to report the absence of rest facilities, and more likely to confirm the lack of dedicated rest facilities and appointed managers. Interestingly, employees in businesses with fewer than 20 people showed contradictory results, not feeling the necessity for appointed managers or the installation of rest facilities.

This study is the first to investigate compliance with standards and awareness of the law's enactment post-legislation. Future research and appropriate system improvements based on the results of this study are hoped for, particularly concerning workplace rest facilities.

## Supporting information

**S1 Questionnaire.**
(DOCX)

## Author Contributions

**Data curation:** Woo-Je Lee.

**Formal analysis:** Woo-Je Lee.

**Investigation:** Yeon-Hee Jeong.

**Supervision:** Ki-Youn Kim.

**Writing – original draft:** Yeon-Hee Jeong.

**Writing – review & editing:** Ki-Youn Kim.

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
