## [Decision Letter · Decision Letter 0]

16 Jan 2024

PONE-D-23-40165Investigation of The Status of Rest Facilities at Industrial Sites and Awareness of Relevant Laws and Regulations of South KoreaPLOS ONE

Dear Dr. Kim,

Thank you for submitting your manuscript to PLOS ONE. After careful consideration, we feel that it has merit but does not fully meet PLOS ONE’s publication criteria as it currently stands. Therefore, we invite you to submit a revised version of the manuscript that addresses the points raised during the review process.

We look forward to receiving your revised manuscript.

Kind regards,

Hamed Aghaei, Ph.D.

Academic Editor

PLOS ONE

3. We note that your Data Availability Statement is currently as follows: [All relevant data are within the manuscript and its Supporting Information files]

Reviewers' comments:

Reviewer's Responses to Questions

**Comments to the Author**

1. Is the manuscript technically sound, and do the data support the conclusions?

Reviewer #1: Partly

Reviewer #2: Yes

2. Has the statistical analysis been performed appropriately and rigorously? 

Reviewer #1: I Don't Know

Reviewer #2: Yes

3. Have the authors made all data underlying the findings in their manuscript fully available?

Reviewer #1: Yes

Reviewer #2: Yes

4. Is the manuscript presented in an intelligible fashion and written in standard English?

Reviewer #1: No

Reviewer #2: No

5. Review Comments to the Author

Reviewer #1: 

Introduction:

- In the third line of the introduction, please mention the full title of the word "OECD", and if necessary, use the abbreviation in the rest of the text.

- In line 18 of the Introduction, the sentence is incomplete and therefore requires rewriting. “Research 19 on the satisfaction with welfare facilities on construction sites (6) indicated that factors such 20 as dining facilities, clothes changing rooms, and rest facilities have a 60.9% impact”

- The sentence in line 24 of the Introduction should be either deleted or placed in a more appropriate location, such as at the beginning of the Introduction “Rest at work plays an important role in building positive 25 relationships and recharging (9).”

2. Method

It seems like the manuscript has some issues with its methodology. It would be helpful to have a more detailed and complete methodology section that can be thoroughly understood and replicated by other researchers.

Survey item

The details regarding the method of determining questionnaire questions, as well as the validity and reliability of the questionnaire, should be added comprehensively.

Results - General and work-related characteristics

"The distribution across age groups from the 20s to the 50s was relatively even, ranging from 15.4% to 38.8%. However, this sentence is a bit unclear and could be rewritten for better clarity."

Reason of rest installation uninstalled

This section is not mentioned in the survey method item. Please include related information by specifying the tool used, along with its validity and reliability.

Discussion

- Line 24, the sentences “Regarding the reasons for the lack of rest facilities, all survey respondents (including those 19 without rest facilities) cited 'lack of a suitable place for rest facilities', 'lack of awareness by 20 the employer', 'cost issues', and 'other (not feeling the necessity, lack of legal knowledge)' in 21 that order.” It seems that the current discussion is not written appropriately and mostly focuses on the musculoskeletal complications caused by the lack of rest facilities. Therefore, this section should be revised from the end of the mentioned sentence to the beginning of the sentences in line 6, which talks about the actions taken by the Korean Ministry of Employment and Labor following a legal amendment. “Following the legal amendment, the Korean Ministry of Employment and Labor has been”

- In the introduction of the manuscript, the existence of laws related to rest facilities in Turkey, England, India, and Australia is mentioned. However, the discussion in the manuscript should also compare the laws related to rest in South Korea with those of the aforementioned countries."

Please rewrite it.

Conclusion:

The conclusion section of a manuscript should ideally summarize the main findings and points of the study. It should reflect on the research questions and objectives, and provide a clear answer to them. Additionally, it should highlight the significant contributions made by the study to the existing literature. Finally, the conclusion should include recommendations for future research. Please rewrite this section completely.

The conclusion section should be summarized and the most important findings and points of the study should be reflected in it.

Reviewer #2: 

Dear PlOS ONE Journal Editor

I appreciate your consideration in submitting the article to me for reviewing under the title " Investigating the implementation status of active risk management in the operating rooms of selected educational hospitals in Iran in 2023- Manuscript #: PONE-D-23-23449 “.

The results of the review of the article is announced as follows:

Introduction:

- Please include the full title of the word "OECD" in the third line of the introduction.

- The paper is linear in reading, even if English should be revised by a native speaker to improve its readability and quality.

Method

The procedure for choosing questionnaire questions, as well as the validity and reliability of the questionnaire, should be thoroughly described.

Results

Reason of rest installation uninstalled

This part is not addressed in the method of survey section; please provide related information by identifying the tool utilized, as well as its validity and reliability.

Discussion

- Please provide a comparison of South Korea's labor laws with other related countries and international standards with an emphasis on increasing workplace productivity

Conclusion:

The conclusion part of a manuscript should summarize the most important results and points of the study. Furthermore, the conclusion should offer recommendations for further study.

6. PLOS authors have the option to publish the peer review history of their article (what does this mean?). If published, this will include your full peer review and any attached files.

Reviewer #1: No

Reviewer #2: No

---

## [Author Response · Author response to Decision Letter 0]

3 Mar 2024

Response to reviewer's #1 comments on revision (This Text is also uploaded at 'Response to reviewer's' file)

The revised texts are marked in red on the manuscript.

1. Add the full term for OECD and use an abbreviated one (OECD) hereafter.

-> Organization for Economic Cooperation and Development was added in the third line of the introduction.

2. The sentence in the 18th line of the introduction is incomplete, so please rewrite it.

-> We revised the sentence in line 18 of the introduction completely. Please refer to the sentence. 

3. Consider deleting the content stating that factors such as dining facilities, changing rooms, and rest areas account for 60.9% of the satisfaction level in construction sites in the research section of the introduction or repositioning it with the positive aspects of rest.

-> We revised to make it smooth by relocating the sentence 'Research on the satisfaction ~ ' in line 19 of the introduction into the beginning of the previous study of Son, Chang-Baek.

4. Provide a detailed explanation of the methodology and add specific details regarding the validity and reliability of the survey items

-> We added the following texts to the method section, detailing the recruitment of subjects and selection of survey items. The reliability analysis results that a Cronbach's α was 0.781, indicating a high reliability had already been mentioned.

- The study subjects were recruited after obtaining consent to participate in the survey anonymously among working people (excluding minors).

- Questionnaire items were created based on the detailed items in the standards for installing and managing rest facilities under Article 194-2 of the Enforcement Regulations of the Occupational Safety And Health Act (see table 1).

5. Make sure to clarify any unclear aspects regarding the even distribution of survey subjects across different age groups.

-> [The distribution across age groups from the 20s to the 50s was relatively even, ranging from 15.4% to 38.8%. However, only one person, accounting for 0.4%, was over the age of 60.]

The above content was revised as follows.

- Looking at the participants in the survey by age group, those in their 20s accounted for 15.4%, those in their 30s accounted for 24.7%, those in their 40s accounted for 38.8%, and those aged 50 and older accounted for 21.1%. (see Table 2)

6. The survey method section did not mention the reasons for not installing rest facilities.

-> The following content was added to the method section.

- For items related to the reasons for not installing rest facilities (Q2-1, Q2-2), the options were selected based on interviews with on-site workers and managers, and the most common opinions were lack of space, cost related issues, lack of awareness by employers, and lack of knowledge about legal matters.

7. There is insufficient discussion regarding the reasons for the lack of rest facilities in the line 24, and the focus seems to be solely on musculoskeletal disorders.

-> We added the texts that no prior research regarding the reasons for not installing rest facilities has been found since the mandatory legislation for rest facility installation took effect in Korea on August 18, 2022. Additionally, all content related to the necessity of rest facilities with a focus on musculoskeletal disorders has been removed.

8. In the introduction, we mentioned the laws related to rest facilities in Turkey, the United Kingdom, India, and Australia, but there is no comparison in the discussion with Korea.

-> We added relevant content to the discussion section. Please refer to it.

9. The conclusion should clearly summarize the findings and key points. It is necessary to emphasize the importance of existing literature and recommend future research directions.

-> The ending part of the conclusion was revised, taking into account the reviewer's comments.

 

Response to reviewer's #2 comments on revision

1. Add the full term for OECD and use an abbreviated one (OECD) hereafter.

-> Organization for Economic Cooperation and Development was added in the third line of the introduction.

2. Provide a detailed explanation of the methodology and add specific details regarding the validity and reliability of the survey items.

-> We added the following texts to the method section, detailing the recruitment of subjects and selection of survey items. The reliability analysis results that a Cronbach's α was 0.781, indicating a high reliability had already been mentioned.

- The study subjects were recruited after obtaining consent to participate in the survey anonymously among working people (excluding minors).

- Questionnaire items were created based on the detailed items in the standards for installing and managing rest facilities under Article 194-2 of the Enforcement Regulations of the Occupational Safety And Health Act (see table 1).

3. The survey method section did not mention the reasons for not installing rest facilities.

-> The following content was added to the method section.

- For items related to the reasons for not installing rest facilities (Q2-1, Q2-2), the options were selected based on interviews with on-site workers and managers, and the most common opinions were lack of space, cost related issues, lack of awareness by employers, and lack of knowledge about legal matters.

4. Please explain Korea’s labor laws in comparison to other countries.

-> A discussion was made by comparing the laws related to rest facilities in Turkey, the UK, India, and Australia mentioned in the introduction with the laws related to rest facilities in Korea.

5. The conclusion should clearly summarize the findings and key points. It is necessary to emphasize the importance of existing literature and recommend future research directions.

-> The ending part of the conclusion was revised, taking into account the reviewer's comments.

---

## [Decision Letter · Decision Letter 1]

10 Apr 2024

Investigation of The Status of Rest Facilities at Industrial Sites and Awareness of Relevant Laws and Regulations of South Korea

PONE-D-23-40165R1

Dear Dr. Kim,

We’re pleased to inform you that your manuscript has been judged scientifically suitable for publication and will be formally accepted for publication once it meets all outstanding technical requirements.

Kind regards,

Hamed Aghaei, Ph.D.

Academic Editor

PLOS ONE

Additional Editor Comments (optional):

Reviewers' comments:

Reviewer's Responses to Questions

**Comments to the Author**

1. If the authors have adequately addressed your comments raised in a previous round of review and you feel that this manuscript is now acceptable for publication, you may indicate that here to bypass the “Comments to the Author” section, enter your conflict of interest statement in the “Confidential to Editor” section, and submit your "Accept" recommendation.

Reviewer #1: All comments have been addressed

Reviewer #2: All comments have been addressed

2. Is the manuscript technically sound, and do the data support the conclusions?

Reviewer #1: Yes

Reviewer #2: Yes

3. Has the statistical analysis been performed appropriately and rigorously? 

Reviewer #1: Yes

Reviewer #2: Yes

4. Have the authors made all data underlying the findings in their manuscript fully available?

Reviewer #1: Yes

Reviewer #2: Yes

5. Is the manuscript presented in an intelligible fashion and written in standard English?

Reviewer #1: Yes

Reviewer #2: Yes

6. Review Comments to the Author

Reviewer #1: (No Response)

Reviewer #2: (No Response)

7. PLOS authors have the option to publish the peer review history of their article (what does this mean?). If published, this will include your full peer review and any attached files.

Reviewer #1: No

Reviewer #2: No
